# Differential Expression of ATM, NF-KB, PINK1 and Foxo3a in Radiation-Induced Basal Cell Carcinoma

**DOI:** 10.3390/ijms24087181

**Published:** 2023-04-13

**Authors:** Rim Jenni, Asma Chikhaoui, Imen Nabouli, Anissa Zaouak, Fatma Khanchel, Houda Hammami-Ghorbel, Houda Yacoub-Youssef

**Affiliations:** 1Laboratory of Biomedical Genomics and Oncogenetics (LR16IPT05), Institut Pasteur de Tunis, University Tunis El Manar, Tunis1002, Tunisia; 2Department of Dermatology, Habib Thameur Hospital (LR12SP03), Medicine Faculty, University Tunis El Manar, Tunis 1008, Tunisia; 3Anatomopathology Department, Habib Thameur Hospital (LR12SP03), Medicine Faculty, University Tunis El Manar, Tunis 1008, Tunisia

**Keywords:** radiotherapy, radiation-induced BCC, low-dose effects, DNA repair, ATM-NF-kb signaling, *PINK1* gene, microRNA, biomarkers

## Abstract

Research in normal tissue radiobiology is in continuous progress to assess cellular response following ionizing radiation exposure especially linked to carcinogenesis risk. This was observed among patients with a history of radiotherapy of the scalp for ringworm who developed basal cell carcinoma (BCC). However, the involved mechanisms remain largely undefined. We performed a gene expression analysis of tumor biopsies and blood of radiation-induced BCC and sporadic patients using reverse transcription-quantitative PCR. Differences across groups were assessed by statistical analysis. Bioinformatic analyses were conducted using miRNet. We showed a significant overexpression of the *FOXO3a*, *ATM*, *P65*, *TNF-α* and *PINK1* genes among radiation-induced BCCs compared to BCCs in sporadic patients. *ATM* expression level was correlated with *FOXO3a*. Based on receiver-operating characteristic curves, the differentially expressed genes could significantly discriminate between the two groups. Nevertheless, *TNF-α* and *PINK1* blood expression showed no statistical differences between BCC groups. Bioinformatic analysis revealed that the candidate genes may represent putative targets for microRNAs in the skin. Our findings may yield clues as to the molecular mechanism involved in radiation-induced BCC, suggesting that deregulation of ATM-NF-kB signaling and *PINK1* gene expression may contribute to BCC radiation carcinogenesis and that the analyzed genes could represent candidate radiation biomarkers associated with radiation-induced BCC.

## 1. Introduction

Basal cell carcinoma (BCC) is the most frequent skin cancer, accounting for 90% of cutaneous cancer, and the most common human malignancy worldwide, characterized by an increasing incidence [1]. According to the Global Cancer Observatory (GLOBOCAN) database,1,198,073 new cases of non-melanoma skin cancer, including BCC, were reported in 2020 [2]. While ultraviolet radiation (UV) is the most known carcinogen involved in skin malignancy, ionizing radiation (IR) exposure is an established etiological risk factor. In fact, radiotherapy is a standard treatment regimen for numerous solid tumors, and it was adopted for the treatment of some benign lesions of the skin, notably ringworm of the scalp (tinea capitis) [3]. Indeed, epidemiological studies have reported that previous depilatory radiotherapy is associated with a fourfold increased risk of cutaneous malignancy in irradiated skin, primarily BCC [4]. BCC occurs after a long period, up to decades after exposure, as a late effect of irradiation, with an inverse correlation between carcinogenesis risk and age at radiation exposure [5,6].

In addition, conflicting data exist concerning the clinical profile and the prognosis of radio-induced BCC compared to BCC in sporadic cases. Although some studies have reported a similar clinical course independently of radiotherapy history, other reports have described a more aggressive response associated with higher recurrence rates and difficult-to-treat radiation-induced BCC [7,8,9].

Aberrant activation of Hedgehog signaling is defined as a key common driver of BCC development. Moreover, high-throughput analyses have highlighted the potential contribution of other signaling pathways in BCC tumorigenesis, notably the WNT, Hippo, NOTCH, and mTOR pathways [10,11]. Nevertheless, little attention has been paid to the underlying molecular mechanisms involved in radiation-induced BCC, and studies that were conducted to identify molecular signatures unique to these tumors are controversial [12,13]. It was suggested that identifying the molecular link between the DDR pathway and Hedgehog signaling could unveil the pathogenesis of BCC developed post-radiotherapy [14].

Furthermore, studies based on genomic and transcriptomic profiling are ongoing to decipher the molecular mechanisms of radiogenic cancer and to identify molecular signatures that are discriminatory between radio-induced cancer and its sporadic counterpart. These were explored notably in radiation-induced hematological malignancies [15], sarcoma [16] and thyroid cancer [17]. While some reports have revealed molecular patterns unique to radiation-induced cancer with a lack of consistency, others have failed to identify these signatures, and the mechanisms underlying radiation-induced carcinogenesis are still largely misunderstood [18,19,20]. In fact, IR induces double-strand DNA breaks (DSBs) which activate ATM protein, a master regulator of the DNA damage response (DDR) pathway, leading subsequently to a cascade of cellular responses that may ultimately contribute to different cell fates [21]. It was reported that different signaling pathways are activated in response to IR in normal and tumor cells, including cellular senescence [22], mitochondrial signaling [23], inflammation [24], oxidative stress response [25], autophagy [26] and microRNA signaling [27,28,29]. Indeed, it was suggested that tumor recurrence following radiotherapy may be explained by radiation-induced cellular senescence of adjacent normal cells surrounding the tumor [22,30,31].

In this study, we aimed to decipher the molecular mechanism(s) of radiation-induced BCC tumorigenesis by investigating the expression patterns of genes of interest in radio-induced BCC and exploring their potential usefulness as candidate biomarkers specific to this tumoral entity. Moreover, we aimed to study the microRNA-mRNA regulatory network to understand the potential contribution of microRNA regulators in this pathogenesis. Our results identified differential gene expression between radiation-induced and non-radiation-induced BCC, suggesting that the deregulation of ATM-NF-kB signaling and *PINK1* expression could be involved in the pathogenesis of radiation-induced BCC.

## 2. Results

### 2.1. Clinical Features of BCC Patients

Patients were separated into two groups depending on history of radiotherapy of the scalp for tinea capitis during their childhood: radiation-induced (radio-induced) BCC and non-radiation-induced (non-radio-induced) BCC developed in sporadic patients.

According to the experience of the referral center, the “Dermatology department of Habib Thameur Hospital”, the majority of patients treated with radiotherapy for tinea capitis develop nodular BCC on the scalp.

To avoid any bias due to patient selection that may have affected gene expression levels, the patients enrolled in both groups were selected to match different parameters (age at BCC diagnosis, UV exposure, etc.) as closely as possible. All collected tumors were localized on the face (six cases) and on the scalp (two cases). All tumors were primary BCCs. Mean age at diagnosis was 66.25 and 65.5 for the radio-induced and non-radio-induced groups, respectively. The tumors were of nodular or adenoid histology independently of radiation etiology. The majority of patients, especially radio-BCC patients, showed multiples lesions. Adenoid tumors coexisted with basosquamous carcinoma at the same anatomical site or with morphea form BCC ata nearby site in the cases of BCC2 and BCC5, respectively. Basosquamous carcinoma and morphea form BCC are aggressive BCC subtypes. Within our cohort, BCC patients presented different clinical phenotypes (according to the numbers and sites of lesions and histopathological subtypes) independently of IR exposure. Clinical and anatomopathological parameters were obtained from medical records and are summarized in Table 1. Different clinical and histopathological BCC subtypes are presented in Figure 1 and Figure 2 respectively.

### 2.2. Molecular Analysis

All tumors located on the head was selected as an inclusion criterion. This decision was made in order to exclude any differential gene expression associated with molecular signatures linked to UV exposure. First, we investigated *FOXO3a*, *ATM*, *P65*, *PINK1* and *TNF-*α expression in radio-induced BCC and non-radio-induced BCC and in age-matched healthy controls using the rt-qPCR technique. Our data revealed significant upregulation of *FOXO3a*, *ATM*, *P65*, *PINK1* and *TNF-α* genes among the radio-induced BCC patients compared to the non-radio-induced BCC group (*p* = 0.021 for each of the *ATM*, *P65*, *PINK1* and *TNF-α* genes; *p* = 0.043 for *FOXO3a*) (Figure 3).

Furthermore, a significant strong positive correlation was observed between *ATM* and *FOXO3a*’ expression among BCC patients using the Pearson correlation test (r = 0.951, *p* = 0.000289) (Figure 4). A positive correlation was also observed between *ATM* and *P65* (r = 0.984, *p* = 0.000009) and between *ATM* and *TNF-α* (r = 0.976, *p* = 0.000036).

We further examined the potential of *FOXO3a*, *ATM*, *P65*, *PINK1* and *TNF-α* genes to discriminate between radio-induced and non-radio-induced BCC using ROC curve analysis. We defined cut-off values of 0.179, 7.956, 7.09, 16.78 and 0.46 for *FOXO3a*, *ATM*, *P65*, *PINK1* and *TNF-α*, respectively, based on the Youden index method. Our results revealed that the gene expression levels could significantly differentiate between BCC groups (area under the curve (AUC) = 1, *p* = 0.021, for the *ATM*, *P65*, *PINK1* and *TNF-α* genes, with 100% specificity and 100% sensitivity, and AUC = 0.938, *p* = 0.043, with 100% sensitivity and 75% specificity, for *FOXO3a*) (Figure 5). In fact, these genes showed an excellent diagnostic performance, which suggests their usefulness as potential diagnostic biomarkers for radio-induced BCC.

Subsequently, the expression levels of the *TNF-α* and *PINK1* genes were analyzed in the blood of patients with BCC to assess their potential as non-invasive biomarkers for radio-induced BCC. Our data showed a similar expression profile for *TNF-α* among BCC patients in both groups (*p* > 0.05). Inaddition, *PINK1* analysis revealed upregulation among radio-induced BCC compared to non-radio-induced BCC, which was in accordance with the *PINK1* expressionin patients’ biopsies. However, this differential expression failed to reach statistical significance (*p* > 0.05) (Figure 6).

### 2.3. Bioinformatic Analysis

We examined putative interactions between the differentially expressed genes and microRNAs (miRNAs) in the skin using the miRNet database. An miRNA-target interaction network was created that included several upstream miRNAs that interact directly with FOXO3a, ATM, P65, TNF-α and PINK1 mRNAs, representing a complex landscape of functional associations. Each node represents an mRNA or miRNA (Figure 7). In fact, miRNAs are intricately involved in the DDR pathway by targeting ATM in the skin. FOXO3a and ATM had the highest interaction (highest connectivity degree) and hence were the most important hubs in this network analysis. miR-24-3p was the most important miRNA regulating the network based on its having the highest node degree (Table 2). According to the Reactome database, cellular senescence was the most enriched pathway (*p* = 0.0026, adj. *p* < 0.05).

The majority of these miRNAs are described in the literature as IR-responsive and are primarily involved in cellular senescence (this will be discussed below). Hence, we suggest that deregulation of these miRNAs in response to IR may affect skin cells’ response to irradiation, notably linked to the analyzed pathways, which may be associated with the carcinogenesis process.

## 3. Discussion

In our study, we aimed to identify differentially expressed genes between radio-induced and non-radio-induced BCC which may provide insights into the molecular contributors to or the under pinning sofradio-induced carcinogenesis. To the best of our knowledge, our study was the first to explore the ATM-NF-kB pathway and *PINK1* expression in radio-induced BCC. In addition, we suggest that miRNA regulation of these pathways may shed light on radio-induced BCC.

BCC represents a significant health and socio-economic burden given its high frequency and incidence [32]. BCC is rarely invasive or metastatic an dis usually associated with a good prognosis. However, it may lead to significant morbidity due to its locally destructive potential and its relatively high recurrence rate [33]. In addition, different BCC entities exist, though the genetic basis of the carcinogenic process and clinical differences are not fully elucidated [34]. Moreover, it is unclear whether mechanisms of BCC carcinogenesis differ according to etiological factors.

IR is a well-known risk factor for skin malignancies. Clinical and epidemiological studies have reported a high rate of BCC cancer following IR exposure, which has primarily been described in the context of treatment for tinea capitis during childhood [4]. It was suggested that cancer risk may persist throughout the life of exposed individuals [35]. Conflicting data exist concerning the prognosis of radio-induced BCC.

-Clinical features of radio-induced BCC and non-radio-induced BCC

In the present study, patients were characterized by different clinical profiles independently of radiation etiology, presenting adenoid or nodular subtypes. Although nodular BCC is the most frequent subtype, adenoid tumors are very rare histopathological variants that are considered low-grade BCCs [36]. Adenoid tumors were isolated or associated with other subtypes, notably basosquamous and morphea form BCC, for BCC2 and 5, respectively, which are aggressive BCC subtypes [37,38]. Nodular BCC was classified as a low-risk tumor according to the WHO classification. According to BCC risk stratification, BCC8 and BCC19 are associated with an intermediate prognosis, as they are characterized as nodular subtypes, with sizes> 2 cm, and are located in low-risk areas [39]. Based on the new BCC classification, tumors may be classified as “easy-to-treat” or “difficult-to-treat”, which classifications are associated with different clinical outcomes [40,41]. Although some tumors included in our study were “easy-to-treat” BCCs, some of them presented specific features that may pose treatment difficulties, which may be linked to tumor location, as with BCC12 and BCC16 sporadic cases (located in patients’ noses).These tumors have a high risk of recurrence. In fact, our data are consistent with a study which reported that IR exposure does not affect clinical and histopathological features or even patient prognosis [42]. However, other studies have reported a more aggressive phenotype among radio-induced BCC [9]. Herein, prognosis was assessed based on clinical prognostic factors.

-Gene expression analysis to decipher radio-induced BCC carcinogenesis mechanisms

Despite radiation exposure being a known carcinogenic factor for BCC, the underlying molecular mechanisms are not yet fully understood. Although those studies have reported the absence of molecular signatures that could discriminate between radio-induced and non-radio-induced BCC, particularly linked to the *PTCH* and *TP53* genes [13,43], it was suggested that the Hh signaling pathway could be linked to susceptibility to BCC radiation carcinogenesis and that *PTCH* could contribute to this process [43]. In fact, it was reported that the genetic background of Ptch-mutated mice may affect susceptibility to BCC development after IR exposure [44,45]. Furthermore, it was suggested that genomic instability could explain genetic susceptibility to radio-induced BCC carcinogenesis [46], which may predispose to mitochondrial DNA mutations [47]. Recently, a genomic characterization of radio-induced BCC versus sporadic cases has revealed specific chromosomal rearrangements that could discriminate between both groups [12].

-ATM-NF-kB signaling and PINK1-mediated mitophagy in response to irradiation

In our study, we report differentially expressed genes between BCC groups that are radiation-responsive and mainly involved in IR-driven BCC carcinogenesis. *ATM* and *FOXO3a* expression were significantly increased in BCC biopsies from irradiated patients compared to sporadic cases. In fact, IR induces DSBs which trigger the DDR pathway. *ATM* is a master regulator involved in the coordination of cellular response to radiation-induced DNA damage and a key determinant of radio-sensitivity [48]. In addition, *ATM* expression was significantly correlated with *FOXO3a* expression level in BCC patients. This suggests a potential regulation, at least in part, of *ATM* by *FOXO3a* transcription factors. Indeed, it was reported that in the DDR pathway, FOXO3a interacts directly with ATM, which is required for ATM autophosphorylation, promoting the ATM response network [49]. However, our findings were in agreement with the fact that FOXO3a regulates *ATM* gene expression, which has been reported in hematopoietic stem cells [50]. Nonetheless, it was described that, in response to DNA damage, *FOXO3a* upregulation maintains genomic stability and is mediated by H2AX, a downstream target of ATM [51,52]. However, *FOXO3a* and *ATM* expression levels were not correlated among three radio-induced patients, which may suggest that other factors are involved in *ATM* regulation in BCC radiation carcinogenesis. Moreover, we revealed upregulation of *P65*, which coded for an NF-kB subunit, and of *TNF-α*, which coded for a pro-inflammatory cytokine and a senescence-associated secretory phenotype (SASP) factor, among irradiated patients. A positive correlation was observed between ATM and P65 and between ATM and *TNF-α*, which could suggest a molecular link between ATM and NF-kB. In fact, IR triggers cellular senescence in normal and cancer cells. ATM is a key driver of NF-kB-induced cellular senescence following DNA damage through the secretion of SASP factors, which enhance NF-kB signaling in an autocrine manner via cytokine receptors [22]. Our data suggest an aberrant activation/over-presentation of the ATM-NF-kB signaling pathway in response to IR. Despite this, we did not directly analyze NF-kb signaling activation. The high expression levels of *TNF-α* and *P65* are indicative markers of NF-kB activation. A few studies have been performed to assess the biological effect of IR on human skin cells which revealed the activation of the DDR pathway, an inflammatory response [53,54,55], and the expression of cellular senescence markers [56]. Moreover, in the context of cancer radiotherapy, IR induced cellular senescence of normal cells adjacent to tumors, leading to either tissue fibrosis and organ dysfunction or to tumorigenesis [22,30,31]. Indeed, it was suggested that transient senescent cells are associated with an anti-tumoral effect, while their accumulation is linked to a detrimental effect leading to increased cancer susceptibility [57]. 

Furthermore, ATM plays an important role in mitochondrial responses to irradiation, following recognition of nuclear DNA damage and subsequent damage signal transduction to mitochondria, playing an important role in the crosstalk between these two organelles in irradiated fibroblast cells, which enhance mitochondrial biogenesis and ROS generation [58]. Although nuclear DNA was primarily considered as a main target of IR, the involvement of mitochondria as key targets of IR and as contributors to radio-induced cell impairment is becoming increasingly evident [23,59] and could explain the high risk of carcinogenesis [60]. Despite the role of mitochondrial dysfunction in response to irradiation being well understood, the involvement of mitochondrial integrity and dynamics has been less explored. In fact, ATM can promote selective elimination of defective mitochondria, termed mitophagy, via activation of the PINK1/Parkin pathway; however, the mechanism involved is not fully elucidated. It was described that ATM impacts this pathway via kinase-dependent [61] or -independent mechanisms through the regulation of *PINK1* (PTEN-induced putative kinase1) expression or interaction with Parkin [62,63]. In addition, ATM-deficient cells revealed that mitophagy induction following irradiation is ATM-dependent [64]. We analyzed *PINK1* gene expression, a key contributor to mitophagy, and observed over expression in radio-induced BCC compared to non-radio-induced cases. In addition, FOXO3a enhances *PINK1* gene expression, leading to a pro-survival signal in response to oxidative stress [65]. This was in accordance with another study which suggested that under extensive mitochondrial alterations, mitophagy activation enhances cell survival following irradiation [66]. However, it was suggested that post-irradiation mitophagy induction could maintain cellular redox control and that disruption of this pathway leads to tumorigenesis [60]. Another report has described that mitophagy effect depends on irradiation dose [67]. 

In summary, our results suggest that the ATM-NF-kB signaling pathway and the *PINK1* gene, related to cellular senescence and mitophagy, are involved in radio-induced BCC carcinogenesis.

-Biomarkers of radio-induced BCC

ROC curve analysis revealed that the differentially expressed genes could significantly discriminate between radio-induced BCC and non-radio-induced BCC and may consequently represent candidate radio-induced BCC biomarkers. This could guide further validation studies aiming to investigate the relevance of these candidate biomarkers with larger cohorts based on independent sets of samples. Such studies could provide straightforward evidence of potential radiation biomarkers and be a step toward understanding radio-susceptibility to BCC. Indeed, accumulating studies are seeking to identify molecular peculiarities unique to radiation-induced cancer compared to sporadic counterparts, as they are often clinically indistinguishable, which may lead to the decipherment of key events involved in radiation carcinogenesis [15]. Conflicting data have been generated, while some reports have identified specific signatures associated with molecular fingerprints of radiation-induced cancer. Others have often described similar molecular patterns independently of radiation etiology, and no accurate molecular signature/biomarker has yet been settled [16,18,68]. Further studies are warranted to untangle the underlying mechanisms. Therefore, we examined whether these biomarkers in particular, *TNF-α* and *PINK1* could serve as non-invasive tools by analyzing their expression profiles in patients’ blood. *TNF-α* was under-expressed in BCC patients, with a similar pattern independent of radiation etiology. These data may support that *TNF-α* alteration was tumor-specific. This finding was supported by a loop between radio-induced inflammation, mobilization of immune cells and sustained tissular damage, mediated by DNA damage and SASP factors following irradiation [69]. However, our results were in discordance with previous studies that reported persistent systemic inflammation in individuals with a history of IR exposure [70]. This was explained by possible confounding factors or by the combinatorial effect of patient age and irradiation [71,72,73]. In fact, the BCC patients included in our study were age-matched; accordingly, the differences in gene expression probably reflect cellular response to irradiation. In addition, inflammatory systemic response may be explained by blood cell response to irradiation, since these individuals were exposed to whole-body rather than localized irradiation. On the other hand, *PINK1* was differentially expressed between the two BCC groups, with overexpression in irradiated cases; however, this differential expression failed to reach statistical significance. In fact, an increase in *PINK1* expression in rat thyroids was reported following irradiation [74]. Our results point out, at least, that the analyzed genes could not represent non-invasive radio-BCC biomarkers.

-miRNA regulators in BCC radiation carcinogenesis

Our current understanding of cellular response to irradiation has evolved to a more complex network that involves different molecular determinants, such as epigenetic regulators, mainly miRNAs [75]. miRNAs mediate cellular radio-response via regulating components of key signaling pathways involved in this process, specifically DDR signaling [76]. Indeed, it was suggested that gene expression profile alterations following irradiation are largely explained by miRNAs, which play a crucial role in defining cell fates, controlling subsequent radiosensitivity [77]. Moreover, it was described that miRNAs participate in radiation carcinogenesis [78,79,80]. In this study, miRNAs represented in the network analysis created by miRNet are mostly described as radio-responsive. miR-24-3p represents the most important miRNA in the generated network, targeting the *ATM*, *RELA* (*P65*) and *PINK1* genes. In fact, it was reported that miR-24-3p plays a crucial role in carcinogenesis and therapeutic response associated with radiosensitivity [81]. miR-24 was described as a regulator of cellular response to DNA damage, and its downregulation enhances tumor cell radio-resistance [82]. Furthermore, miR-24 expression decreases with cellular senescence [83], and it was defined as a mitochondrial miR (mitomiR) [84]. It was described that miR-24-3p and miR-124 attenuate the expression of pro-inflammatory mediators via NF-kB signaling [85,86]. miR-124 also increases in senescent skin, while it decreases during tumorigenesis, particularly in squamous cell carcinoma [87]. In addition, it was reported that Let-7 family expression patterns are altered upon irradiation, characterized by ATM-dependent downregulation in skin fibroblasts [88]. miR-26a and miR-100a were described as enhancing tumor cell radio-sensitivity via targeting DNA repair proteins [89,90]. miR-145 is described as a regulator of the DDR pathway and cellular senescence in different cell types [91,92]. Downregulation of miR-145-5p was also reported in basal cell carcinoma [93]. Indeed, it was revealed that senescence-associated miRNAs (SA-miRs) are implicated in radiation-induced premature cellular senescence and that the knockdown of miR-155 stimulates this process [94]. miR-181a/b are involved in keratinocyte replicative senescence and are dysregulated in cancers [95]. According to miRNA transcriptome profiling, miR-222 was repressed in irradiated cells [96]. Moreover, miR-24 is a radiation-responsive miRNA that may represent a potential biomarker of radiation-induced gastric cancer [97]. We suggest that this network may give insights into the regulatory role of microRNAs in radio-induced BCC; some of these miRNAs may contribute to the pathogenesis of BCC through the regulation of ATM-NF-kB signaling and PINK1 expression level in response to irradiation. Experimental studies are warranted to validate these interactions in radio-induced BCC.

Further research in the field of normal tissue radiobiology is of paramount importance to decipher the mechanisms involved in cellular radio-response. In fact, even though depilatory radiotherapy is no longer used, IR is still used as a clinical diagnostic tool and a therapeutic modality, besides being used for occupationally/accidentally exposed individuals, and the incidence of BCC is continuously increasing. Moreover, patients with cancers, including some BCCs amenable to radiotherapy, may develop a second malignancy due to the radiosensitivity of healthy tissues surrounding the tumor, which affects treatment efficiency. The identification of promising biomarkers is an unmet need to improve long-term risk assessment following IR exposure, which may aid clinicians in monitoring exposed individuals. These biomarkers may represent novel targets to prevent or at least minimize cellular radiotoxicity linked to carcinogenesis risk.

To our knowledge, our study represents the first investigation of ATM-NF-kB signaling and *PINK1* gene expression, implicated in cellular senescence and mitophagy in BCC pathogenesis following radiotherapy. Further studies are needed to deeply characterize and explore these processes in a larger cohort of radio-induced BCC patients. Moreover, proteins encoded by the studied genes should be analyzed to further confirm the activation of these pathways. In addition, further studies are mandatory to shed light on the complexity of the interactions between genes of interest and microRNAs in response to irradiation.

## 4. Materials and Methods

### 4.1. Human Specimens and Sample Collection

We conducted a comparative study using patients’ fresh frozen skin biospecimens and blood. According to histological evaluation, confirmed by an atomopathologist, a total of eight patients diagnosed with basal cell carcinoma were enrolled in this study. Four of the BCC samples were obtained from sporadic patients and four were obtained from patients with a history of radiotherapy of the scalp for tinea capitis during their childhood and were considered as radio-induced BCC. All tumors being located on the face and scalp area was an inclusion criterion. One healthy skin biopsy from a UV-exposed area from the head of a healthy age- and geographically matched control was included after screening for the absence of any signs of malignancy. Systemic investigation was performed for five of the patients, including three radio-induced BCCs and two BCCs from sporadic cases (or non-radiation-induced BCC), and the sample from the healthy control. BCC and healthy control biopsies were collected from the Dermatology department of “Habib Thameur” Hospital.

All participants included in this study provided written informed consent, and the study was carried out in accordance with the Helsinki principles and approved by the Institute Pasteur Ethics Committee in Tunisia under the ethical accord number (reference PCI/22/2012/v2). 

### 4.2. Gene Expression Analysis

#### 4.2.1. RNA Extraction and cDNA Synthesis

Total RNA from the tissue and blood samples of patients with BCC and the healthy control were extracted using trizol and the miRNeasy Mini Kit (QIAGEN), following the manufacturer’s instructions. The concentration and purity of isolated RNA was assessed using the Nanodrop spectrophotometer DeNovix DS-11 (Thermo Fisher Scientific, Wilmington, DE, USA).

Prior to total isolated RNA reverse transcription (RT), a Dnase treatment was carried out in a Qsp of 10 µL, using 1 µg of RNA, 10X DNase Buffer and 1 U/µL of DNase I enzyme. After RNA sample incubation at ambient temperature for 15 min, 25 mM of EDTA was added, and RNA was then placed at 65 °C for 10 min. The cDNA synthesis was performed using the Superscript II RT Kit (Cat. No:18064014 Invitrogen, Carlsbad, CA, USA), starting with 1µg of purified RNA in a total reaction volume of 20 µL, according to the manufacturer’s protocol.

#### 4.2.2. Quantitative Real-Time PCR

The *ATM*, *FOXO3a*, *P65* (*RELA*), *TNF-α* and *PINK1* genes were investigated by quantitative real-time PCR (qPCR) using the Syber Green kit, according to the manufacturer’s protocol. Among genes involved in cellular senescence and mitophagy, this set of genes was selected after exploring gene expression data generated by the GTEx (Genotype-Tissue Expression) database. These genes are expressed in human skin cells and showed no differences between sun-exposed and non-sun-exposed skin areas. Specific primers were used. The qPCR reaction was carried out in a 96-well plate using the Light Cycler L480 (Roche Applied System) with a reaction volume containing Sybr Green Mix (Invitrogen), Quantities of 10 µM of each forward and reverse primer for 17 µL of Mix and 100 ng/µL of cDNA were used. cDNA amplification was performed in a cycling protocol with 35 cycles (at 95 °C for 15 s and for 60 °C for 1 min) preceded by a pre-amplification step at 95 °C for 10 min. Relative gene expression levels were normalized to the expression of the housekeeping gene *RPLP0*. Experiments were performedin duplicate for each candidate target gene and for each reference gene. Relative fold changes were calculated using the comparative cycle threshold (ct) method (2^−ΔΔCt^). The sequences of the primers are listed in Table 3.

### 4.3. Statistical Analysis

IBM SPSS version 21.0 was used for statistical analysis. Differences across specimen groups and gene expression correlations were inspected using the Mann–Whitney U and Pearson exact tests, respectively. Area under the curve–receiver operating characteristic (AUC-ROC) analysis was performed to evaluate the potential of candidate genes to discriminate between radio-induced BCC and non-radio-induced BCC groups. An excellent model for a good separation performance was considered when the AUC value was near to 1. Cut-off values were determined according to the Youden index method. Maximizing the Youden index enabled the identification of the cut-off point of the curve with the highest value for the sum of sensitivity and specificity. For all tests, a *p*-value < 0.05 was considered statistically significant.

### 4.4. Bioinformatic Analysis

We explored the miRNet 2.0 database to identify putative upstream microRNAs that target the candidate genes and generate an miRNA-target gene interaction network-based visual analysis of the skin. In fact, node degrees (the numbers of connections with other nodes) indicate important hubs in the generated network. In addition, miRNet offers a functional enrichment analysis, which was performed with the Reactome pathway tool (adjusted *p* < 0.05). 

## 5. Conclusions

We suggest that aberrant regulation of ATM-NF-kB signaling and *PINK1* gene expression could be related to BCC radiation carcinogenesis and that biomarkers involved in these pathways may discriminate between BCCs according to their radiation etiologies. Furthermore, we suggest that microRNAs may play an important role as well, highlighting their potential implication in the pathogenesis of radio-induced BCC. Our study highlights the importance of the exploration of radio-induced carcinogenesis and extends our understanding to a more complex network involving different interconnected cellular and molecular components. 

## Figures and Tables

**Figure 1 ijms-24-07181-f001:**
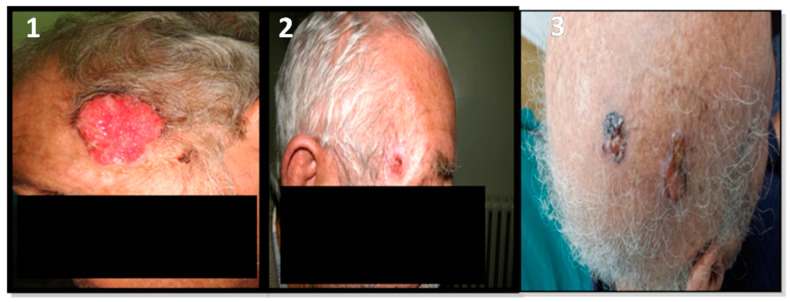
Radio-induced and non-radio-induced BCC patients. 1-corresponds to a non-radio-induced BCC patient. 2,3-correspond to two radio-induced patients: 2-presents one small malignant lesion; 3-presents multiple lesions on the scalp. The aggressive form of BCC is independent of radiation etiology in this cohort.

**Figure 2 ijms-24-07181-f002:**
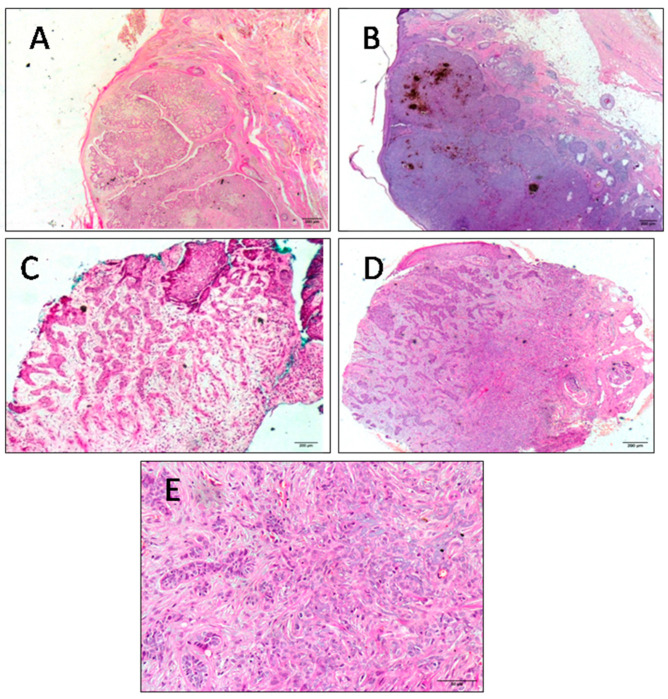
Hematoxylin-and-eosin-stained section of BCC tumor tissues representative of different histopathological subtypes of BCC. (**A**) Adenoid basal cell carcinoma with pseudo-glandular appearance. The stroma is fibrous (×100). (**B**) Nodular subtype of basal cell carcinoma with melanin deposit (×100). (**C**) Morphea form basal cell carcinoma with narrow cords of basaloid cells which are compressed by abundant sclerotic collagenous stroma (×100). (**D**) Basosquamous basal cell carcinoma (×100). (**E**) A focal squamous differentiation (×200). Scale bars = 200 µm for (**A**–**D**) and 50 µm for (**E**).

**Figure 3 ijms-24-07181-f003:**
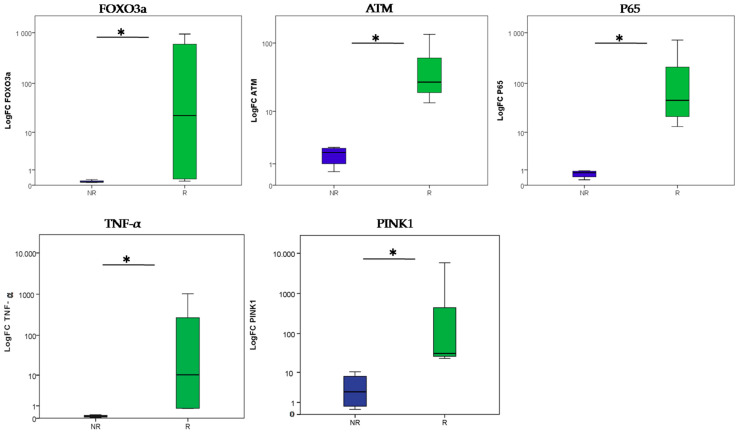
Differential gene expression profiles for radio-induced and non-radio-induced BCC biopsies. Blue box plot: non-radio-induced BCC (NR); green box plot: radio-induced BCC (R). (* = *p* < 0.05).

**Figure 4 ijms-24-07181-f004:**
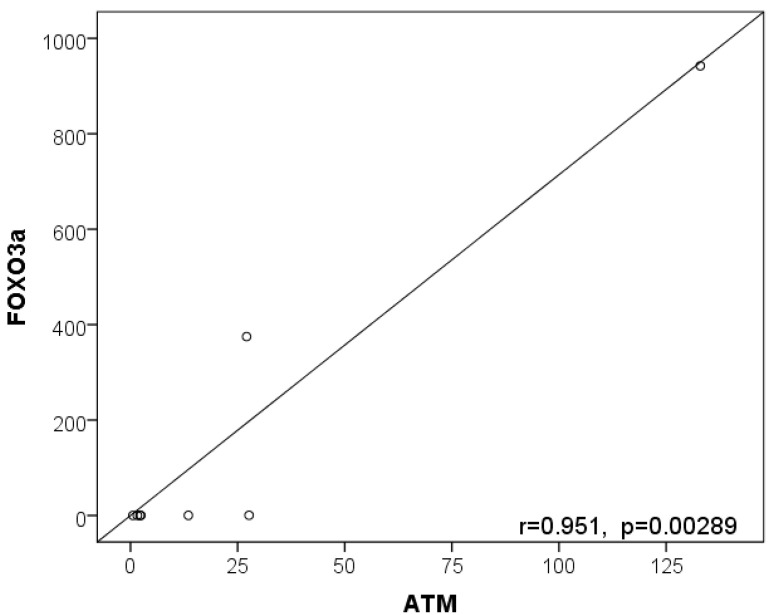
Correlation between *FOXO3a* and *ATM* expression in BCC patients’ biopsies. Pearson correlation of *FOXO3a* and *ATM* gene expression fold changes among BCC patients revealed a strong positive correlation of these genes with a Pearson correlation coefficient (r) = 0.951 (*p* = 0.00289). Each circle represents the expression levels of both genes for each patient.

**Figure 5 ijms-24-07181-f005:**
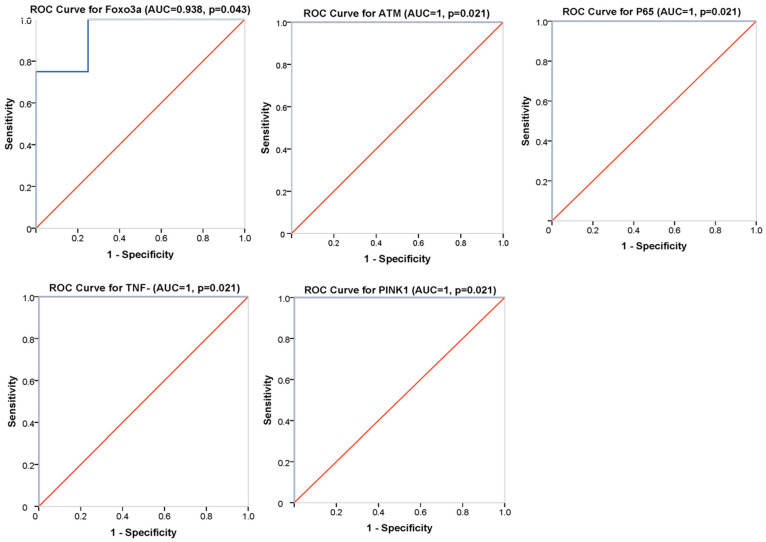
ROC curve analysis of differentially expressed genes between radio-induced and non-radio-induced BCC. ROC analysis for the *FOXO3a*, *ATM*, *P65*, *PINK1* and *TNF-α* genes in different plots. The *p*-value is the probability that the observed sample AUC-ROC is found when the true AUC-ROC is 0.5 (null hypothesis: the variable cannot distinguish between the two groups; area = 0.5 if the ROC coincides with the reference line). We found that the area under the curve is significantly different from 0.5. *ATM*, *P65*, *PINK1* and *TNF-α* have excellent discrimination performance, with AUCs = 1 (100%specificity and 100% sensitivity). The ROC curve foreach of them passes through the upper left corner of the plot, while *FOXO3a* has an ROC curve close to the upper left corner, with an AUC = 0.938 that represents an excellent diagnostic value.

**Figure 6 ijms-24-07181-f006:**
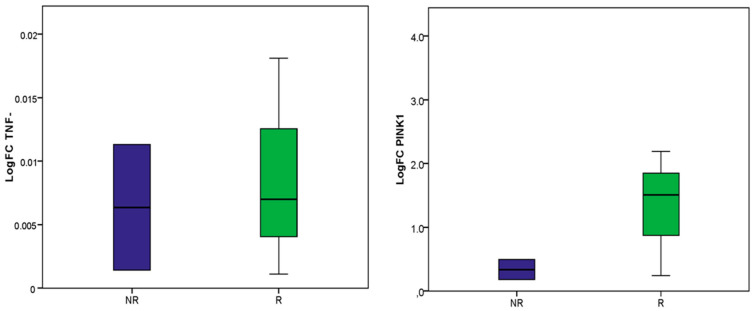
Differential gene expression profiles between radio-induced and non-radio-induced BCC in patients’ blood. Blue box plot: non-radio-induced BCC NR; green box plot: radio-induced BCC R.

**Figure 7 ijms-24-07181-f007:**
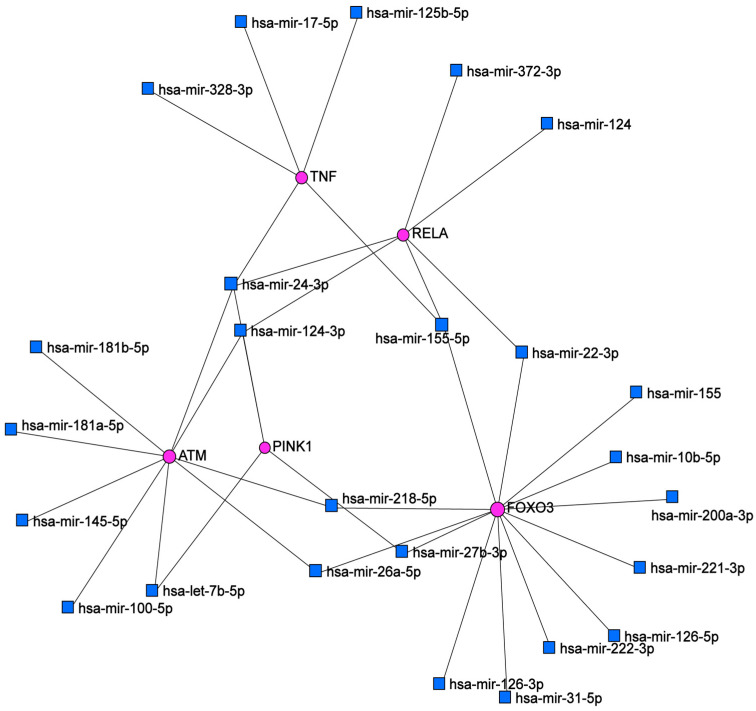
miRNA-mRNA interaction network with differentially expressed genes in the skin, constructed using miRNet database data.

**Table 1 ijms-24-07181-t001:** Patients’ characteristics.

Patient	Gender	Ethnicity	Age	History of Radiotherapy	Histology	Tumor Size	Multifocality	Tumor Location
BCC2	M	Tunisia	60	Yes	Adenoid	3 cm	Yes (n = 7)	Temporal region of the face
BCC4	M	Tunisia	60	Yes	Adenoid	3 cm	Yes (n = 2)	Temporal region of the face
BCC5	M	Tunisia	79	No	Adenoid	1 cm	Yes (n = 3)	Temporal region of the face
BCC8	M	Tunisia	85	Yes	Nodular	2.5 cm	No	Scalp
BCC12	F	Tunisia	63	No	Nodular	3 cm	Yes (n = 2)	Ala of nose
BCC16	M	Tunisia	40	No	Nodular	3 cm	No	Ala of nose
BCC19	M	Tunisia	80	No	Nodular	3 cm	No	Cheek
BCC22	M	Tunisia	60	Yes	Nodular	2 cm	Yes (n = 2)	Scalp

**Table 2 ijms-24-07181-t002:** Top miRNAs and targeted mRNAs in the network interactions determined via miRNet.

Node	Degree
FOXO3a	13
ATM	9
RELA	6
TNF	5
hsa-mir-24-3p	4
PINK1	4
hsa-mir-124-3p	3
hsa-mir-155-5p	3
hsa-mir-26a-5p	2
hsa-mir218-5p	2
hsa-mir-27b-3p	2
hsa-mir-22b-3p	2
hsa-mir-22b	2
hsa-mir-let-7b-5p	3

**Table 3 ijms-24-07181-t003:** Primer sequences used for qPCR.

Gene	Sequence (5’-3’)	Length	Tm (°C)
*FOXO3a*	F: CGGACAAACGGCTCACTCT R: GGACCCGCATGAATCGACTAT	19 21	61.9 61.7
*ATM*	F: GCACGAAGTGCCTCCAATTC R: ACATTCTGGCACGCTTTGG	21 19	61.1 61.4
*TNF-α*	F: CCTCTCTCTAATCAGCCCTCTG R: GAGGACCTGGGAGTAGATGAG	22 21	62.1 62.8
*PINK1*	F: CCCAAGCAACTAGCCCCTC R: GGCAGCACATCAGGGTAGTC	19 20	64.5 63.1
*P65*	F: ATGTGGAGATCATTGAGCAGC R: CCTGGTCCTGTGTAGCCATT	21 20	60 60.2

## Data Availability

All data have been provided in the manuscript. The data generated in this study can be provided by the corresponding author upon reasonable request.

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
