# Peer review of "Differential Expression of ATM, NF-KB, PINK1 and Foxo3a in Radiation-Induced Basal Cell Carcinoma"

_ijms, 2023, doi:10.3390/ijms24087181_

Round 1
Reviewer 1 Report (New Reviewer)
The manuscript “Gene expression patterns and miR-mRNA network analysis in radiation-induced Basal cell carcinoma” by Jenni R. et al., aims to investigate the molecular mechanism involved in radiation-induced BCC tumorigenesis. To reach their goal, they examined the gene expression pattern of 6 tumoral samples by qRT-PCR looking for potential biomarkers candidates for this tumor type. The manuscript is well written and this type of investigation can significantly contribute to the field. However, looking at the analysis and results presented by the authors, it is premature to take any conclusion and more experiments are necessary. Here are some of my major concerns about it:
-
Only 6 tumoral samples are not enough to take any conclusion about tumoral biomarkers.
-
Table 1 is completely out of the layout and needs to be fixed. It is also missing the “Age” on the table`s header.
-
What is the protein level of the differential expressed genes? I would recommend the authors check the protein level by Western Blot.
-
I strongly recommend checking the RNA expression by Northern Blot.
-
In Figure 5, the ROC curve is not showing any information. The curves are not displayed on the graphic.
Author Response
The manuscript “Gene expression patterns and miR-mRNA network analysis in radiation-induced Basal cell carcinoma” by Jenni R. et al., aims to investigate the molecular mechanism involved in radiation-induced BCC tumorigenesis. To reach their goal, they examined the gene expression pattern of 6 tumoral samples by qRT-PCR looking for potential biomarkers’ candidates for this tumor type. The manuscript is well written and this type of investigation can significantly contribute to the field. However, looking at the analysis and results presented by the authors, it is premature to take any conclusion and more experiments are necessary. Here are some of my major concerns about it:
Authors: We would like to thank the reviewer 1 for his comments. We tried to address your comments one by one, and we hope that the text has been improved accordingly.
Comment1: Only 6 tumoral samples are not enough to take any conclusion about tumoral biomarkers.
Response1: We would like to thank the reviewer for his comment. In fact, it is a pilot study that focus on specific genes that are involved in cellular senescence and mitophagy (selected based on RNAseq database; as we have described in “Material and Methods” section in the manuscript L439-441) and some of them have been reported in the literature as relevant radiation biomarkers/ involved in response to radiotherapy (Le Reun et al.,2022; Wang, Z et al.,2020).
We agree with the reviewer, we were limited by the number of patients (n=8 not 6) obtained from our collaborating hospital. In fact, radiation-induced BCC is not a frequent tumor. In addition, we have selected a set of patients to match as closely as possible to different parameters (mainly linked to tumor site) to avoid any modifying factor leading to a bias in the results’ interpretation of genes’ expression.
In addition, based on ROC curve analysis, we have reported that analysed genes may represent candidate biomarkers (L321-323). This could guide further validation studies aimed to explore in depth these genes and to investigate the relevance of these candidate biomarkers on a larger cohort based on independent set of samples. We have added this information in the discussion section in the manuscript to (L323-325).
Comment2: Table 1 is completely out of the layout and needs to be fixed. It is also missing the “Age” on the table`s header.
Response 2: We apologize for the quality of the Table1. We have improved it as required.
Comment3: What is the protein level of the differential expressed genes? I would recommend the authors check the protein level by Western Blot.
Response3: We would like to thank the Reviewer1 for this important suggestion, and we agree that Western blot analysis could complement and strengthen our findings. Unfortunately, tumor biopsies were small and they were entirely used to perform RNA expression analysis. Further research are needed to explore these genes at a protein level (as we mentioned in the discussion section L402-404).
Comment4: I strongly recommend checking the RNA expression by Northern Blot.
Response4: We thank the reviewer for his suggestion. Unfortunately, Northern blot analysis could not be performed due to the lack of biological samples. In fact, given the small quantities of mRNA that we obtained following RNA extraction, we opted to use rt-qPCR analysis. Although, Northern blot (NB) is an accurate technique for RNA quantification, qPCR is much more sensitive than NB, less time consuming and does not require large amounts of RNA. This allows for rapid and reliable quantification of mRNAs of interests. In the present study, gene expression analysis was performed in duplicate (for each candidate gene and for reference gene) to ensure the validity of qPCR results.
Comment5: In Figure 5, the ROC curve is not showing any information. The curves are not displayed on the graphic.
Response5: In Figure 5, curves were represented in color code on the same graphic. In fact, FOXO3a gene has an AUC of 0.938, was represented in blue. Nevertheless, each of ATM, P65, PINK1 and TNF-α, present an AUC of 1. Therefore, the ROC curve of each of them pass through the upper left corner of the plot, thus, they are superimposed curves.
The graphic has been modified to avoid confusion; each curve is now displayed separately.

Reviewer 2 Report (New Reviewer)
I have read and evaluated the manuscript entitled “Gene expression patterns and miR-mRNA network analysis in radiation-induced Basal cell carcinoma” based on the results of qPCR analysis of ATM, NF-kB, and PINK1 genes and statistical analyses in basal cell carcinoma but several issues that makes the article unsuitable for publication of the current form.
Major:
1. Data for microRNAs are poor, so the title should be reconsidered along the results. Maybe focus on differential expression of ATM, NF-kB, and PINK1 in BCC samples.
2. The authors consider that the molecular mechanism for radiation-induced BCC suggests that the deregulation of ATM-NF-kB signaling and PINK1 expression may contribute to BCC radiation carcinogenesis and could be a candidate of radiation biomarker, however, I am not sure ATM-NF-kB signaling are suggested in this context, because, e.g., no data for phosphorylation of ATM protein and ubiquitination of IKB protein even in cell culture experiments and biochemistries in vitro in this study. All text should be based on the fact.
3. It is better present fold changes in Figs 3 and 6.
4. Unbalanced expression patterns of ATM and FOXO3a in the three patients, in Fig 4, need to be discussed.
5. If the authors strongly claim the miRNA-mRNA network in radiation-induced BCC, at least, expression of target miRNAs and these binding to 3’-UTR of TNF, RELA, ATM, and FOXO3 genes must be shown in luc assay or EMSA.
Minor:
1. Add scale bar in each panel in Fig2a-e.
2. Node maybe better sorted by either degree or betweenness in Table 2.
Author Response
Responses to reviewer 2:
Reviewer 2: I have read and evaluated the manuscript entitled “Gene expression patterns and miR-mRNA network analysis in radiation-induced Basal cell carcinoma” based on the results of qPCR analysis of ATM, NF-kB, and PINK1 genes and statistical analyses in basal cell carcinoma but several issues that makes the article unsuitable for publication of the current form.
Authors: We would like to thank the reviewer 2 for his precious time in reviewing our manuscript and providing valuable comments. We tried our best to address your comments one by one and we hope that the text is improved accordingly.
Major:
Comment1: Data for microRNAs are poor, so the title should be reconsidered along the results. Maybe focus on differential expression of ATM, NF-kB, and PINK1 in BCC samples.
Response1: We have modified the title of the manuscript as required, from: "Gene expression patterns and miR-mRNA network analysis in radiation-induced Basal cell carcinoma" to "Differential expression of ATM, NF-KB, PINK1 and Foxo3a in radiation-induced basal cell carcinoma"
Comment2: The authors consider that the molecular mechanism for radiation-induced BCC suggests that the deregulation of ATM-NF-kB signaling and PINK1 expression may contribute to BCC radiation carcinogenesis and could be a candidate of radiation biomarker, however, I am not sure ATM-NF-kB signaling are suggested in this context, because, e.g., no data for phosphorylation of ATM protein and ubiquitination of IKB protein even in cell culture experiments and biochemistries in vitro in this study. All text should be based on the fact.
Respone2: We would like to thank the reviewer for his relevant comment. We totally agree that other experiments should be performed to analyze the ATM-NF-kB pathway, notably by analyzing proteins encoded by these genes, which we pointed out in the discussion section of the manuscript (L402-404). However, this could not be carried out because of the unavailability of appropriate biological samples. In fact, the biopsies which were small in size, were entirely used for RNA expression analysis.
Moreover, ATM-NF-Kb signaling has been reported as an important driver of radiation-induced cellular senescence in normal cells (Li, M et al.,2018). It was described that ATM is a key mediator of DNA damage-induced NF-Kb mediated cellular senescence, and that inhibition of ATM reduces P65 level, NF-Kb activation and the expression of senescence associated secretory phenotype secretion (SASP) (including TNF- α) in a mice model (Zhao,J et al.,2020).
In the present study, the upregulation of P65, coding for an NF-kB subunit, and of TNF-α could reflect the activation of NF-kB signaling. In addition, a correlation between ATM, P65 and TNF-α genes expression was determined by Pearson test (data not shown) which may suggest the molecular link between ATM and NF-kB, which could reflect the over-representation of ATM-NF-kB signaling. We added this information in the text to lines (L142-L143) and (L276-278). However, as you have mentioned, further studies are mandatory to explore this pathway in BCC radiation carcinogenesis.
Comment3: It is better present fold changes in Figs 3 and 6.
Response3: We would like to thank the reviewer for his recommendation. However, radiation-induced group represents a very low levels of genes of interest. Representing their expression on fold change mask/obscure the distribution and show an expression level close to zero, especially given the great difference of gene levels between radiation-induced and non-radiation induced groups.
Comment4: Unbalanced expression patterns of ATM and FOXO3a in the three patients, in Fig 4, need to be discussed.
Response4: We thank the reviewer for his comment. Figure3, presents the correlation between ATM and FOXO3a expression levels among the eight patients. The four overlapping circles that are localized on the line, each corresponding to a non-radiation-induced BCC patient. They represent a close expression level of FOXO3a and ATM. However, the three patients who do not show an association between FOXO3a and ATM levels, are radiation-induced BCC, which may indicate that other factors are involved in the regulation of ATM gene expression in these patients. This is consistent with the fact that ATM signaling is over-represented after exposure to ionizing radiation. We have discussed this point in Line 272-274.
Comment5: If the authors strongly claim the miRNA-mRNA network in radiation-induced BCC, at least, expression of target miRNAs and these binding to 3’-UTR of TNF, RELA, ATM, and FOXO3 genes must be shown in luc assay or EMSA.
Response5: We would like to thank the reviewer for his comment. We concur with the reviewer that other experiments should be performed, nonetheless, the proposed techniques are unavailable in our laboratory. In fact, a computational analysis was conducted in the present study using an in-silico tool (miRNet), to predict putative interactions between miR and mRNA encoded by analyzed genes, based on predicted 3'UTR mRNA complementarity with microRNA "seed region". As we have mentioned in the discussion section, those identified microRNAs are described in the literature as radio-responsive (L361-362) and have been involved in the carcinogenesis process, which may suggest that identified microRNA-mRNA network may be associated with radiation-induced BCC. However, further studies (notably, experimental studies) are warranted to validate these interactions and its contribution to BCC radiation carcinogenesis. We have pointed out this point in the discussion to line L386-387.
Minor:
Comment1: Add scale bar in each panel in Fig2a-e.
Response1: Scale bar are added to the Figure2 as required.
Comment2: Node maybe better sorted by either degree or betweenness in Table 2.
Response2: The table 2 is modified as recommended, it represents nodes degree. Betweenness is removed from the table.
References:
|
Wang, Z., Chen, B., Liao, K., Xu, H., Xing, J., Wei, Z., ... & Ren, L. (2020). Suppression of FoxO3a Increases Responsiveness of Glioma Cells to Radiation Treatment. Le Reun, E., Bodgi, L., Granzotto, A., Sonzogni, L., Ferlazzo, M. L., Al-Choboq, J., El-Nachef, L., Restier-Verlet, J., Berthel, E., Devic, C., Bouchet, A., Bourguignon, M., & Foray, N. (2022). Quantitative Correlations between Radiosensitivity Biomarkers Show That the ATM Protein Kinase Is Strongly Involved in the Radiotoxicities Observed after Radiotherapy. International journal of molecular sciences, 23(18), 10434. |
|
|
Li, M., You, L., Xue, J., & Lu, Y. (2018). Ionizing radiation-induced cellular senescence in normal, non-transformed cells and the involved DNA damage response: a mini review. Frontiers in pharmacology, 9, 522. |
|
|
Zhao, J., Zhang, L., Lu, A., Han, Y., Colangelo, D., Bukata, C., ... & Robbins, P. D. (2020). ATM is a key driver of NF-κB-dependent DNA-damage-induced senescence, stem cell dysfunction and aging. Aging (Albany NY), 12(6), 4688.
|
|

Round 2
Reviewer 1 Report (New Reviewer)
The authors properly answered all the questions and the provided modifications improved the manuscript`s quality. Please revise for minor grammatical and mistyping errors.
Reviewer 2 Report (New Reviewer)
The authors sufficiently respond to the reviewer’s comments.
This manuscript is a resubmission of an earlier submission. The following is a list of the peer review reports and author responses from that submission.
Round 1
Reviewer 1 Report
Re: ijms-2129685
Title: Cellular senescence and mitophagy in radiation-induced basal cell carcinoma developed after childhood radiotherapy of the scalp
This study aims to find genes which were differentially expressed in therapy-related radiation-induced basal cell carcinoma in comparisons of sporadic BCC with biopsied tissues and blood samples. Authors suggested five candidate molecules including FOXO3a, ATM, P65, TNF-α and PINK1 genes as significantly overexpressed in radiation-induced BCC by bioinformatic analysis, then concluded the involvement of cellular senescence and mitophagy in the pathogenesis of radiation-induced BCC.
Major points
This paper is seriously suffered from a shortness of experimental design and a misinterpretation of their results. English language should be edited by a native speaker.
Results
Line 106: What’s meaning of “different BCC clinical phenotypes”.
Lines 103-105: Subtypes of BCC should be shown by more common term such as basosquamous and morpheaform (sclerosing, morphoeic), and should be shown as histological pictures.
Table 1 should include age, age at radiation, estimated radiation dose, gender, ethnicity, and other complicated malignancy. The number of multifocality of BCC5 and right side of Table 1 are missing.
Lines 116-117: The head is sun-exposed area. Why did authors think it being possible to exclude the link of UV exposure.
Lines 123-124, 131-134: These are discussion.
Lines 140-147: How about cut-off values of gene expression to differentiate? Authors should validate their results with another set of cases.
Discussion
Lines 214-233: These are about pathological features of BCC in this study associated with prognosis. No meaningful to characterize radiation-induced BCC.
Lines 247-298: These are too speculative to discuss with present results. Authors should carefully clarify details about DDR-, pro-inflammatory, and senescence-associated pathway.
Lines 301-331: These are also too speculative to discuss with present results.
Materials and methods
Line 460: Authors should describe why they focused on five genes as candidate molecules for biomarker of radiation-induced BCC.
There are no descriptions about validation analysis.
Author Response
Responses to reviewer1
Reviewer 1 :
This study aims to find genes that were differentially expressed in therapy-related radiation-induced basal cell carcinoma in comparisons of sporadic BCC with biopsied tissues and blood samples. Authors suggested five candidate molecules including FOXO3a, ATM, P65, TNF-α and PINK1 genes as significantly overexpressed in radiation-induced BCC by bioinformatic analysis, then concluded the involvement of cellular senescence and mitophagy in the pathogenesis of radiation-induced BCC.
Comment 1: Major points
This paper is seriously suffered from a shortness of experimental design and a misinterpretation of their results. English language should be edited by a native speaker.
Response1: We would like to thank the reviewer for his comment. Unfortunately, we were limited in our experiments by the lack of biological samples, obtained by the collaborating hospital. As radiation-induced BCC is not a frequent tumor entity, and given that long-term patient follow-up has only been provided by a few hospitals, including our collaborator, patients’ recruitment was quite difficult. Hence the scarcity of samples and the shortness of experimental design, which we pointed out in the discussion section of the manuscript (L441-443).
- Regarding the English language, we apologize for the errors in the manuscript. The typos and grammar mistakes were revised, and the article has been proofread by a native English speaker.
Results
Comment 2: Line 106: What’s meaning of “different BCC clinical phenotypes”.
Response 2: Sorry for this confusion. BCC patients present different clinical features defined as different clinical phenotypes (which include number, site and histopathological subtype of BCC tumors), the term was also mentioned by Madan V et al.,2006 [1]. We modified the sentence in the text as follows: BCC patients present different clinical phenotypes (according to number, site and histopathological subtype) (L109-110).
Comment 3: Lines 103-105: Subtypes of BCC should be shown by more common term such as basosquamous and morpheaform (sclerosing, morphoeic), and should be shown as histological pictures.
Response 3: We thank the reviewer for his relevant suggestion. We changed terms as proposed: sclerodermiform was replaced by morpheaform, and metatypical was replaced by basosquamous carcinoma (L105-108, and in the discussion section L232). Histological pictures were added to the text as required (Figure 2).
Comment 4: Table 1 should include age, age at radiation, estimated radiation dose, gender, ethnicity, and other complicated malignancy. The number of multifocality of BCC5 and right side of Table 1 are missing.
Response 4: Age at diagnosis, gender and ethnicity were added as requested to the Table1. For radiation dose, patients have no information concerning this issue. In fact, the dose and modality of irradiation could not be precisely established/evaluated. In Tunisia, depilatory radiotherapy was used between 1922-1963 and the protocol that was adopted is of Kienbock-Adamson method (mean dose: 450-850rad per field)[2]. Furthermore, patients indicate that they received radiotherapy of the scalp during their childhood without precise the age at time of irradiation. There is no other complicated malignancy among included patients.
Comment 5: Lines 116-117: The head is sun-exposed area. Why did authors think it being possible to exclude the link of UV exposure.
Response 5: In fact, ultraviolet (UV) and ionizing radiation are both risk factors for BCC. All BCC tumors are located in continuously sun-exposed area (defined as an inclusion criteria), in this case, patients differ according to ionizing radiation exposure (radio-induced and non-radio-induced BCC groups). Therefore, all patients may express genes linked to UV exposure, however the difference of gene expression level detected between the two groups is attributable to radiation etiology.
Comment 6: Lines 123-124, 131-134: These are discussion.
Response 6: We would like to thank the reviewer for pointing this out. Sentences L123-124 were moved to L348-349 in the discussion section. The sentences L133-134 were moved to L270-271 in the discussion section as well.
Comment7: Lines 140-147: How about cut-off values of gene expression to differentiate? Authors should validate their results with another set of cases.
Response7: The optimal cut-off values were determined based on the Youden index method (the most commonly used method, which is defined at a cut-off point as the sum of the associated sensitivity and specificity minus one or as the difference between sensitivity and the false positive rate.
Maximizing the Youden index is able to identify the cut point of the curve that has the highest value in the sum of sensitivity and specificity. We have revised the Material and Methods section and added this information to Line 503-505.Cut-off values were: 0.179, 7.956, 7.09, 16.78 and 0.46 for FOXO3a, ATM, P65, PINK1 and TNF-alpha, respectively. We have mentioned this in L153-154.
Unfortunately, we were unable to validate on a larger set of patients, given the lack of biological samples.
Discussion
Comment 8: Lines 214-233: These are about pathological features of BCC in this study associated with prognosis. No meaningful to characterize radiation-induced BCC.
Response 8: We would like to thank the reviewer for this comment. Indeed, we have described pathological features of BCC based on prognostic clinical factors and we have reported different clinical course (linked to recurrence risk) independently of radiation exposure. The aim was only to describe that BCC patients’ stratification in different risk groups is not linked to radiation etiology. We have changed the subtitle from: "Clinical features of radio-induced BCC compared to non-radio-induced BCC" to "Clinical features of radio-induced BCC and non-radio-induced BCC (L225).
Comment 9: Lines 247-298: These are too speculative to discuss with present results. Authors should carefully clarify details about DDR-, pro-inflammatory, and senescence-associated pathway.
Response 9: We agree that this part is speculative at this time. Thus, we have edited this section and removed 4 paragraph L294-L315)
Comment 10: Lines 301-331: These are also too speculative to discuss with present results.
Response10: We would like to thank the reviewer to point this out, we agree with his comment. The paragraph has been modified accordingly to make this part more objective and clearer (L331-L349).
Materials and methods
Comment 11: Line 460: Authors should describe why they focused on five genes as candidate molecules for biomarker of radiation-induced BCC.
Response 11: In fact, we have explored gene expression data from skin generated by GTEx (Genotype-Tissue Expression) database. We have selected a set of genes involved in cellular senescence and mitophagy, that are expressed in human skin cells and that showed an absence of significant difference between sun exposed and non-sun exposed skin area, this includes ATM, FOXO3a, PINK1, P65 (RELA), TNF-α genes. We have revised the text and clarify this point L475-481.
Comment 12: There are no descriptions about validation analysis.
Response 12: We thank the reviewer1 for this significant comment. In fact, we have highlighted genes that could represent candidate biomarkers for radiation-induced BCC. these genes will guide future research aimed at exploring in depth the processes of cellular senescence and mitophagy in the pathogenesis of BCC. Further validation studies on a larger cohort would be necessary. We mentioned it in L441-L443; L444-L446.
References
- Madan, V., et al., Genetics and risk factors for basal cell carcinoma. British Journal of Dermatology, 2006. 154: p. 5-7.
- Khalfat, A., RESULTATS ET CRITIQUES DES TRAITEMENTS MASSIFS DE LA TEIGNE. International Journal of Dermatology, 1967. 6(1): p. 38-41.

Reviewer 2 Report
Comment on:
Cellular senescence and mitophagy in radiation-induced basal cell carcinoma developed after childhood radiotherapy of the scalp, by Rim Jenni, Asma Chikhaoui, Imen Nabouli, Anissa Zaouak, Houda Hammami-Ghorbel, Houda Yacoub-Youssef.
This paper addresses a very interesting subject of radiobiology research, that is, the role of senescence and mitophagy in the pathophysiology of radiation-induced basal cell skin carcinoma (BCC). A small series of BCC patients undergoing scalp irradiation for tinea capitis during their childhood, was compared with cases of sporadic, non-radio-induced BCC, matched for age, age at diagnosis of BCC, sex, UV exposition, lesion’s diameter (1-3cm), and also nodular or adenoid histology type independently from a radiation history. A molecular analysis (rt-qPCR) addressed FOXO3a, ATM, P65, PINK1, TNF-, revealing their significant upregulation in irradiated patients, thus suggesting the activation of the ATM-NF-B-TNF- pathway and mitophagy in this group. A significant strong correlation between ATM and FOXO3a was shown using the Pearson correlation test, indicating FOXO3a as a potential transcriptional regulator of ATM. Very high level of sensitivity and specificity (AUC = 0.938-1) were obtained for FOXO3a, ATM, P65, PINK1, TNF-, at the ROC analysis, despite the smallness of the patient series. Conversely, the gene analysis on circulating blood failed to show significant differential expression between the two groups. However, a putative bio-informatic analysis was performed on the miRNAs interacting with the above genes, depicting a network of functional interactions whose degrees of connectivity with p65 (RELA), ATM, PINK1, FOXO3 genes, are significantly coherent with the enrichment of the cellular senescence pathway, according to the Reactome database.
These original disclosures may represent clues to the molecular drivers underlying the ionizing radiation-induced skin carcinogenesis, as widely dealt with in the section 3. (Discussion) of this paper.
Although positively evaluating the originality and scientific relevance of this contribution, this reviewer raises some criticisms.
1) He wonders at the lack of mitophagy and senescence histochemical and/or fluorescence microscopy images (see as references, e.g., respectively: Dolman NJ et Al., Autophagy 9;11:1653-62 (2013); Debacq-Chainiaux F et Al., Nature Protocols 4;12:1794-806 (2009), to be related to the other findings. These microscopic images could strengthen the conjectural evidence of the already reported data in favor of the theoretical framework provided by these authors.
2) Figure 3 and 4 are poorly readable: their quality should be improved (e.g.: symbols’ size, line’s thickness and colors).
3) Although the section 3. (Discussion) is remarkably long (and sometimes redundant in the opinion of this reviewer) a point should be shortly addressed regarding the Hedgehog pathway (probably not related to the subject here dealt with) deserving a remarkable interest in the current literature as a possible mechanism for enhanced radiation susceptibility in developing skin BCC with particular aggressive behavior. However, the genetic analysis of p53 and PTCH genes in human BCCs revealed no differences between irradiated and non-irradiated patients (Tassone A et Al., Aesthetic Plastic Surg 36:1387-92 (2012)). Contrarily, some previous experimental evidence on animal models has suggested that the genetic background may contribute to the modification of Ptch-associated susceptibility to radiation-induced BCC development (revised in Li C, Athar M, et Al., 36;1387-92 (2012)).
Author Response
- Responses to reviewer2
Reviewer 2:
This paper addresses a very interesting subject of radiobiology research, that is, the role of senescence and mitophagy in the pathophysiology of radiation-induced basal cell skin carcinoma (BCC). A small series of BCC patients undergoing scalp irradiation for tinea capitis during their childhood, was compared with cases of sporadic, non-radio-induced BCC, matched for age, age at diagnosis of BCC, sex, UV exposition, lesion’s diameter (1-3cm), and also nodular or adenoid histology type independently from a radiation history. A molecular analysis (rt-qPCR) addressed FOXO3a, ATM, P65, PINK1, TNF-, revealing their significant upregulation in irradiated patients, thus suggesting the activation of the ATM-NF-B-TNF- pathway and mitophagy in this group. A significant strong correlation between ATM and FOXO3a was shown using the Pearson correlation test, indicating FOXO3a as a potential transcriptional regulator of ATM. Very high level of sensitivity and specificity (AUC = 0.938-1) were obtained for FOXO3a, ATM, P65, PINK1, TNF-, at the ROC analysis, despite the smallness of the patient series. Conversely, the gene analysis on circulating blood failed to show significant differential expression between the two groups. However, a putative bio-informatic analysis was performed on the miRNAs interacting with the above genes, depicting a network of functional interactions whose degrees of connectivity with p65 (RELA), ATM, PINK1, FOXO3 genes, are significantly coherent with the enrichment of the cellular senescence pathway, according to the Reactome database.
These original disclosures may represent clues to the molecular drivers underlying the ionizing radiation-induced skin carcinogenesis, as widely dealt with in the section 3. (Discussion) of this paper.
Although positively evaluating the originality and scientific relevance of this contribution, this reviewer raises some criticisms.
Response : We thank the reviewer 2 for his succinct summary of our work and for recognizing its potential importance. We have revised the manuscript based on these comments.
Comment 1: He wonders at the lack of mitophagy and senescence histochemical and/or fluorescence microscopy images (see as references, e.g., respectively: Dolman NJ et Al., Autophagy 9;11:1653-62 (2013); Debacq-Chainiaux F et Al., Nature Protocols 4;12:1794-806 (2009), to be related to the other findings. These microscopic images could strengthen the conjectural evidence of the already reported data in favor of the theoretical framework provided by these authors.
Response 1: We would like to thank the reviewer2 for this significant comment. We completely agree with your idea. Although that qPCR is a high specific method, it is noteworthy that suggested experiments are interesting and could complement our results. Unfortunately, it could not be performed for lack of biological samples. Indeed, the biopsies, which were small in size, were entirely used for RNA extractions to carry out gene expression study. Further studies are warranted to more explore these processes in radio-induced BCC pathogenesis.
Comment2: Figure 3 and 4 are poorly readable: their quality should be improved (e.g.: symbols’ size, line’s thickness and colours).
Response 2: We apologize for the quality of these figures. We have improved them as required.
Comment3: Although the section 3. (Discussion) is remarkably long (and sometimes redundant in the opinion of this reviewer) a point should be shortly addressed regarding the Hedgehog pathway (probably not related to the subject here dealt with) deserving a remarkable interest in the current literature as a possible mechanism for enhanced radiation susceptibility in developing skin BCC with particular aggressive behavior. However, the genetic analysis of p53 and PTCH genes in human BCCs revealed no differences between irradiated and non-irradiated patients (Tassone A et Al., Aesthetic Plastic Surg 36:1387-92 (2012)). Contrarily, some previous experimental evidence on animal models has suggested that the genetic background may contribute to the modification of Ptch-associated susceptibility to radiation-induced BCC development (revised in Li C, Athar M, et Al., 36;1387-92 (2012)).
Response3: We have omitted some sections from discussion to make it more coherent. We thank the reviewer for this suggestion and for pointing out these works. We have added further details concerning susceptibility to radiation-induced BCC (L251-257). Further explorations are needed to more understand the effect of genetic background on BCC radiation carcinogenesis.

Round 2
Reviewer 2 Report
While positively evaluating the overall revision provided in the corrected version of the paper, this reviewer whould greatly appreciate reading the definitve text after the corrections already accomplished. This will be more easily readable. Infact, some minor grammatical oversights are still present.